# Tiliroside Combined with Anti-MUC1 Monoclonal Antibody as Promising Anti-Cancer Strategy in AGS Cancer Cells

**DOI:** 10.3390/ijms241713036

**Published:** 2023-08-22

**Authors:** Iwona Radziejewska, Katarzyna Supruniuk, Katarzyna Jakimiuk, Michał Tomczyk, Anna Bielawska, Anna Galicka

**Affiliations:** 1Department of Medical Chemistry, Medical University of Białystok, ul. Mickiewicza 2a, 15-222 Białystok, Poland; anna.galicka@umb.edu.pl; 2Department of Medical Biology and Genetics, Medical University of Gdańsk, ul. Dębinki 1, 80-211 Gdańsk, Poland; katarzyna.supruniuk@gumed.edu.pl; 3Department of Pharmacognosy, Faculty of Pharmacy with the Division of Laboratory Medicine, Medical University of Białystok, ul. Mickiewicza 2a, 15-230 Białystok, Poland; katarzyna.jakimiuk@umb.edu.pl (K.J.); michal.tomczyk@umb.edu.pl (M.T.); 4Department of Biotechnology, Medical University of Białystok, ul. Kilińskiego 1, 15-089 Białystok, Poland; aniabiel@umb.edu.pl

**Keywords:** tiliroside, gastric cancer, glycoforms, MUC1, TACAs

## Abstract

Specific changes in mucin-type O-glycosylation are common for many cancers, including gastric ones. The most typical alterations include incomplete synthesis of glycan structures, enhanced expression of truncated O-glycans (Tn, T antigens and their sialylated forms), and overexpression of fucosylation. Such altered glycans influence many cellular activities promoting cancer development. Tiliroside is a glycosidic dietary flavonoid with pharmacological properties, including anti-cancer. In this study, we aim to assess the effect of the combined action of anti-MUC1 and tiliroside on some cancer-related factors in AGS gastric cancer cells. Cancer cells were treated with 40, 80, and 160 µM tiliroside, 5 µg/mL anti-MUC1, and flavonoid together with mAb. Real-Time PCR, ELISA, and Western blotting were applied to examine MUC1 expression, specific, tumor-associated antigens, enzymes taking part in their formation, Gal-3, Akt, and NF-κB. MUC1 expression was significantly reduced by mAb action. The combined action of anti-MUC1 and tiliroside was more effective in comparison with monotherapy in the case of C1GalT1, ST3GalT1, FUT4, Gal-3, NF-κB, Akt mRNAs, and Tn antigen, as well as sialyl T antigen expression. The results of our study indicate that applied combined therapy may be a promising anti-gastric cancer strategy.

## 1. Introduction

Gastric cancer (GC) is stated to be the fifth most common cancer and the third most frequent cause of cancer deaths worldwide [1]. Surgery still remains the main treatment for early-diagnosed, operable patients. For advanced GC, new therapeutic methods are currently being studied [2].

It is reported that many natural plant-based chemical compounds possess anti-cancer properties. They are able to impair and even kill cancer cells without being toxic to healthy cells [3,4,5]. The main group of such compounds is polyphenols, widely distributed in the plant kingdom. They may reveal their anti-cancer effects through different mechanisms, including the regulation of cell cycle signaling, elimination of anti-cancer agents, anti-oxidant enzyme activity, apoptosis, and cell cycle arrest [6,7]. There are clinical trials that attempt to correlate polyphenolic intake with the prevention of specific cancers or the diminished recurrence of cancer after the consumption of flavonoids or food rich in them [8]. The representative of such compounds is tiliroside, glycosidic dietary flavonoid (kaempferol 3-*O*-β-D-(6″-E-*p*-coumaroyl)-glucopyranoside) (Figure 1) [9]. Many pharmacological properties have been reported for tiliroside, including anti-diabetic, anti-inflammatory, anti-microbial, anti-oxidant, anti-viral, as well as anti-cancer [10,11,12,13,14,15,16]. The regulation of MAPK/JNK/p38 axis, NF-κB signaling pathway, and apoptosis induction have been reported in leukemia cells as proposed mechanisms of tiliroside action [17].

Carbohydrates are considered the most abundant complex molecules, which play crucial roles in a variety of cellular interactions involved in physiological and pathological processes [18]. One of the main glycoproteins of epithelial cells is MUC1 mucin, which consists of two subunits, a heavily glycosylated N-terminal extracellular domain, and an intracellular C-terminal domain. Both subunits are linked through hydrogen bonds. In some conditions, the extracellular domain can be released upon the action of proper enzymes [19]. The N-terminal part of MUC1 acts as a cell barrier, participates in cell–cell and cell–extracellular matrix interactions, and can also be involved in signaling pathways connected with transformations related to tumor progression [19,20,21]. The function of the MUC1 C terminal domain is more relevant in signal transduction. It participates in EMT (epithelial-mesenchymal transition) and epigenetic reprogramming, which promotes cancer progression, and can lead to drug resistance and immune evasion [19,22]. MUC1 is commonly overexpressed and reveals specifically altered glycosylation in tumor cells in comparison with normal cells. Mucin glycosylation aberrations influence a variety of cellular activities, including growth, differentiation, and transformation, as well as adhesion, interactions with other cell surface receptors, invasion, and immune surveillance [23,24]. In cancers, the glycan chains are often truncated, and they commonly express tumor-associated carbohydrate antigens (TACAs), such as Tn, T antigens and their sialylated forms (sTn, sT), and also fucosylated Lewis antigens. Such altered glycosylation occurring in tumor cells often results from aberrant expression of specific glycosyltransferases and their substrates that regulate their activity [25,26]. This unique expression of tumor-associated carbohydrate antigens on cancer cells enabled the noticing of them by the National Institutes of Health as significant biomarkers of cancer prognosis [27].

Recently, we have established *p*-coumaric acid, kaempferol, astragalin, and tiliroside as potential anti-cancer agents in AGS gastric cancer cells [28]. We have also demonstrated promising results considering the combined action of rosmarinic acid and anti-MUC1 monoclonal antibody in the same cancer cells [29]. These results prompted us to examine the effects of the action of tiliroside combined with an anti-MUC1 monoclonal antibody in AGS cancer cells. We especially focused on the influence of these agents on cancer cell glycoforms.

## 2. Results

### 2.1. Effects of Tiliroside and Anti-MUC1 mAb on the Viability of AGS Gastric Cancer Cells

According to the previous studies performed with AGS gastric cancer cells, we have established IC_50_ for anti-MUC1 as below 5 µg/mL [29] and for tiliroside as far less than 160 µM [28]. In the present study, we decided to apply 5 µg/mL concentration of anti-MUC1 monoclonal antibody and 40, 80, and 160 µM concentrations of tiliroside.

### 2.2. Effects of Anti-MUC1 mAb on MUC1 Expression

To check the effect of anti-MUC1 monoclonal antibody action on MUC1, we assessed the expression of MUC1 mRNA as well as the expression of this mucin in cell lysates and culture medium. In Figure 2A, it can be seen that MUC1 mRNA has been significantly suppressed by anti-MUC1 mAb. Anti-MUC1 also inhibited the expression of MUC1 cytoplasmic tail, MUC1 in cell lysates, and MUC1 extracellular domain released to the culture medium (Figure 2B, Figure 2C, and Figure 2D, respectively).

### 2.3. Effects of Tiliroside and Anti-MUC1 mAb on Tn Antigen Expression

VVA lectin detected the presence of Tn antigen in cell lysates and the culture medium. The results of the ELISA test showed the suppression of Tn antigen expression in cell lysates by 160 µM tiliroside as well as by anti-MUC1 and mAb together with all concentrations of flavonoid. In the culture medium, a slight inhibitory effect was seen for only 160 µM tiliroside and 80 µM of flavonoid together with mAb (Figure 3).

### 2.4. Effects of Tiliroside and Anti-MUC1 mAb on T, sialyl T antigens Expression, and Enzymes Responsible for Their Formation

Core 1 β1,3-galactosyltransferase (C1GalT1) enables the formation of T antigen (Galβ1,3-GalNAc-O-Ser/Thr). In Figure 4A, the strong inhibitory effect of anti-MUC1 and mAb combined with all used concentrations of tiliroside on C1GalT1 mRNA expression is presented. In the case of the C1GalT1 protein, such an effect was revealed only by 160 µM tiliroside and 40 µM flavonoid combined with anti-MUC1 (Figure 4B). PNA is lectin with a high affinity to T antigen. The ELISA test with this lectin revealed only a slight suppression of T antigen expression in cell lysates by the lowest concentration of flavonoid. In the culture medium, the inhibitory effect is seen only after anti-MUC1 action (Figure 4C).

Gal α2,3-sialyltransferase (ST3GalT1) is responsible for sialyl T antigen formation. Anti-MUC1 and mAb combined with tiliroside action decreased ST3GalT1 expression (Figure 5A). The inhibitory effect of tiliroside, mAb, and antibody combined with 40 and 80 µM of flavonoid on ST3GalIV protein expression is presented in Figure 5B. MAAII lectin (with affinity to SAα2,3-Gal; sialyl T antigen), applied in the ELISA test, enabled the detection of the inhibitory effect of anti-MUC1 as well as mAb with tiliroside action on sialyl T antigen expression in cell lysates. Slight suppression of this antigen was also demonstrated in the culture medium by the action of 80 and 160 µM tiliroside, mAb, and mAb combined with the flavonoid (Figure 5C).

### 2.5. Effects of Tiliroside and Anti-MUC1 mAb on Fucosylated Antigen Expression and the Enzyme Responsible for Its Formation

α1,3-fucosyltransferase (FUT4) is the enzyme responsible for the Fucα1,3-GalNAc structure formation. FUT4 mRNA was significantly diminished by all applied combinations of compounds. The effect was enhanced by the combined action of mAb and tiliroside (Figure 6A). FUT 4 protein expression was inhibited by tiliroside alone and combined with anti-MUC1 but not by mAb used without the flavonoid (Figure 6B). The level of inhibitory effect was similar after the action of the flavonoid separately and with mAb. LTA lectin revealed the level of fucosylated antigen (Fucα1,3-GalNAc) in cell lysates and culture medium. The inhibitory effect was observed for only 40 µM tiliroside and 160 µM flavonoid combined with anti-MUC1 in cell lysates. In the culture medium, suppression was seen for mAb and mAb applied with all concentrations of tiliroside (Figure 6C).

### 2.6. Effects of Tiliroside and Anti-MUC1 mAb on Galectin-3 Expression

Galectin-3 (Gal-3), a β-galactose-binding lectin, is a multifunctional protein associated with the development of many cancers. Tiliroside applied alone, anti-MUC1, as well as flavonoid combined with mAb, significantly inhibited Gal-3 mRNA. Tiliroside used with anti-MUC1 revealed an enhanced effect in comparison to the flavonoid used separately (Figure 7A). In contrast, the applied compounds did not influence Gal-3 protein expression (Figure 7B).

### 2.7. Effects of Tiliroside and Anti-MUC1 mAb on NF-κB Expression

The nuclear factor κB (NF-κB) regulates the expression of genes involved in many processes of cancer development. NF-κB mRNA expression, assessed in quantitative Real-Time PCR, was inhibited by 160 µM tiliroside, mAb, as well as by tiliroside together with anti-MUC1, with intensified effect after combined action (Figure 8A). NF-κB protein was revealed in active form (50 kDa) and as a precursor (120 kDa). Anti-MUC1 did not affect NF-κB. The precursor form was inhibited by tiliroside used separately and in combination with anti-MUC1 without enhancing the effect after being joined with mAb. Active NF-κB expression was decreased by 80 and 160 µM tiliroside and 160 µM flavonoid with anti-MUC1 (Figure 8B).

### 2.8. Effects of Tiliroside and Anti-MUC1 mAb on Akt Expression

Akt kinases are signaling molecules participating in cell growth and differentiation with increased expression in many cancers. Anti-MUC1 and tiliroside combined with mAb inhibited Akt mRNA expression. Surprisingly, 160 µM flavonoid increased the level of Akt expression (Figure 9).

## 3. Discussion

Tiliroside is an example of a flavonoid with potential for anti-cancer therapy [12,13,14,15,16]. Among others, it revealed its anti-cancer properties by acting as a CAXII inhibitor in liver cancer [12] and by reducing the E2F/caspase-3 axis in breast cancer cell carcinoma [13]. Recently we have demonstrated the anti-tumor potential of tiliroside in AGS gastric cancer cells by affecting MUC1 and some specific glycoforms involved in cancer development [28]. It is well established that glycans present on the cell surfaces and extracellular molecules are involved in many vital molecular processes, including cell–cell interaction, cell–matrix adhesion, and various signal transduction cascades, all of them with a key role in cancer development [24,30]. Glycosylation in cancers is specifically altered, truncated-O-glycans, branched-N-glycans, diverse fucosylated, and sialylated terminal antigens are overexpressed [18,23,24,30,31]. Such alterations can be due to changes in expression levels of glycosyltransferases and glycosidases, their localization within the secretory pathway, as well as metabolic alterations and donor substrate availability [31,32,33,34]. The possibility of distinguishing the glycosylation pattern of proteins between healthy and cancer patients creates an excellent opportunity in cancer treatment strategies. One of the main glycoproteins of gastric cancer cells is epithelial MUC1 mucin, a well-known oncoprotein, specifically because of its overexpression and altered glycosylation [23,24,25,26,27,35,36]. Some studies have reported that elevated MUC1 expression is directly associated with a higher risk of invasion and poor prognosis, contributing to the angiogenesis, tumor growth, and development of metastasis [35,37]. Several mechanisms have been proposed for such actions. The loss of cell polarity in cancer cells and, accordingly, the disarray of components of cellular membranes results in the movement of MUC1 from the apical surface to the whole membrane surface, which, successively, causes the colocalization of mucin with other transmembrane receptors, as well as ECM components, which were not available before. Cancerous MUC1 becomes more accessible for interactions with receptors, promoting the induction of invasion, migration, metastasis, or the inhibition of apoptosis [19,20,35,38]. Thus, blocking MUC1 with monoclonal antibodies can inhibit the interaction of mucin with neighboring receptors and suppress the ensuing induction of proper cellular signaling pathways, leading to cancer development [35,36].

In our study, we decided to assess the efficacy of the combined action of anti-MUC1 monoclonal antibody and tiliroside in AGS gastric cancer cells. Our concept was partially based on the promising results of our experiments considering the combined action of anti-MUC1 and rosmarinic acid in these cells [29]. Previously, we also examined the influence of tiliroside (as well as other polyphenolic compounds) on some glycoforms in AGS cancer cells [28]. We assumed that anti-MUC1 could decrease MUC1 expression and, after that, the action of the tiliroside could be enhanced. In this work, we proved that monoclonal antibody action reduced MUC1 expression on the mRNA level as well as MUC1 cytoplasmic tail and extracellular domain expression in cell lysates and the culture medium. Such a conclusion could be in accordance with the conception of Birrer et al., who stated that the internalization of the antibody-binding complex occurs when an antibody binds to its ligand [39]. A similar idea about mAb action was proposed by Wu et al. in a pancreatic cancer model where anti-MUC1 induced the translocation of MUC1 into the cytoplasm and, in this way, inactivation of MUC1 oncogenic signaling was accomplished [40]. Recently, the rationality of applying the anti-MUC1 antibody has also been proven by Merikhian et al. on mouse mammary tumors [35].

In the next steps of our study, we examined the influence of the combined action of anti-MUC1 and tiliroside on some specific glycosyltransferases and carbohydrate antigens formed in the participation of these enzymes. The examined sugar structures, Tn, T, sialyl T, and fucosylated antigens belong to the group of tumor-associated carbohydrate antigens (TACAs), which are evidently marked in a large number of cancers, but not in normal cells. They support tumor cell invasion, cause metastasis, can be involved in evasion of the immune system, and are considered biomarkers for cancer detection [26,41]. Such antigens can be implicated in tumor progression in different ways. Tn antigen overexpression has been reported to stimulate cell proliferation, decrease apoptosis, and increase adhesion and migration in colorectal cancer [42]. T antigen is crucial in the adhesion of many different cancer cells to the endothelium through interaction with galectin-3, and in this way, promotes metastasis [43]. The overexpressed, sialylated form of this antigen in cancer cells has been reported to provide favorable conditions for tumor dissemination, as well as being implicated in evading the immune system and was correlated with more invasive and metastatic phenotypes of prostate and gastrointestinal cancer cells [44,45,46]. Fucosylated epitopes frequently occur on cancer cell surfaces and are associated with different aspects of cancer progression, such as increased cell survival and proliferation, tissue invasion and metastasis, angiogenesis, or multidrug resistance [47]. Our experiments revealed that FUT4 (responsible for Fucα1,3-GalNAc antigen formation) was inhibited by tiliroside action. The effect was enhanced by the combined action of anti-MUC1 with a flavonoid. In the case of C1GalT1 (responsible for T antigen formation) and ST3GalT1 (responsible for sialylated T structure formation), such effects were observed only after the connected action of mAb and a flavonoid. However, the expression of C1GalT1, ST3GalIV, and FUT4 on protein level was inhibited but not intensified by the combined action of anti-MUC1 and tiliroside. In the case of C1GalT1, we observe no clear correlation between mRNA and WB results. For 160 µM tiliroside, we even see opposite results (increased mRNA expression and decreased protein expression in relation to untreated control). We could speculate that such discrepancies are because, e.g., post-translational regulations of proteins or a kind of negative feedback (a high level of protein decreases mRNA expression). To support the outcomes for enzymes, we also determined the expression of proper carbohydrate antigens. Inhibitory effects were observed for Tn, T, and sialyl T antigen in cell lysates and for sialyl T and fucosylated epitopes in the culture medium after combined action of anti-MUC1/tiliroside. As all examined carbohydrate structures are implicated in cancer development, we state that the revealed inhibitory effects after the applied therapy suggest the rationality of such strategies in cancer treatment.

To support such ideas about the benefits of the potential application of anti-MUC1/tiliroside, we decided to also study some other factors involved in cancer development. One of them is galectin-3, a lectin with a high affinity for *β*-galactosides, including T antigen. It was reported that this galectin promotes interactions between tumor and endothelial cells, cell aggregation, angiogenesis, and tumor metastasis. Elevated levels of galectin-3 in cancer cells correlate with immune suppression and progressive tumor stages [48,49]. There were many reports demonstrating that interactions between Gal-3 with MUC1 via T antigen are able to induce various key steps in cancer progression and metastasis [41,50]. Thus, decreasing the expression of the factors involved in such interactions seems to be a promising therapeutic strategy. The results of our experiments revealed that the application of tiliroside and anti-MUC1 inhibited the galectin-3 expression on an mRNA level. The effect was enhanced when combined treatment was used. Upon this result, we can conclude that such action supports the idea mentioned above about the rationality of the application of the proposed therapy. In addition, it has been reported that galectin-3 could promote tumor metastasis primarily in AKT-dependent signaling pathways and could act as a crucial regulator in the development and metastasis of many cancers [51]. Thus, the inhibition of Akt mRNA revealed in our study by anti-MUC1/tiliroside treatment additionally supports the conception of utilizing the presented combined therapy.

The last agent assessed in our work was NF-κB, a transcription factor that is responsible for the upregulation or suppression of many genes, promoting carcinogenesis, and due to this, is recommended as a therapeutic target [52]. Recently, it has been demonstrated that the extracellular domain of MUC1 is able to activate NF-κB [35]. Moreover, it has been reported that this factor uses the MUC1 cytoplasmic part as an adaptor protein or co-transcription factor to regulate the expression of oncogenes leading to inflammation, proliferation, induction of EMT, and epigenetic reprogramming [53,54]. By inhibition of NF-κB using the applied therapy on a mRNA level, we proved the rationality of the proposed therapy. In the case of NF-κB expression, combining mAb with tiliroside clearly enhanced the effect; thus, such a combination seems to be an especially attractive strategy in cancer treatment.

## 4. Materials and Methods

### 4.1. Cell Culture

CRL-1739 human gastric adenocarcinoma cells (AGS) were purchased from American Type Culture Collection (ATCC, Manassas, VA, USA). They were cultured in F-12 medium (Gibco, Waltham, MA, USA) in a humidified atmosphere of 5% CO_2_ at 37 °C. The medium was supplemented with 100 μg/mL streptomycin, 100 U/mL penicillin (Sigma, St. Luis, MO, USA), and 10% Fetal Bovine Serum (FBS) (Gibco, Waltham, MA, USA). Then the cells were seeded into 6-well plates (about 5 × 10^5^ cells/well) and cultured for 24 h in 1 mL of growth medium (FBS-free) supplemented with 40, 80, and 160 μM of tiliroside.

Tiliroside (kaempferol 3-*O*-β-D-(6″-E-*p*-coumaroyl)-glucopyranoside) was isolated from leaves of *Rubus caesius* L. (Rosaceae), and its chemical structure was characterized by using spectral (UV, NMR, MS) methods. The tiliroside was >96% pure, as measured by HPLC [55]. Monoclonal anti-MUC1 antibody was used at a concentration of 5 μg/mL, and tiliroside was combined with anti-MUC1 (40 µM tiliroside/5 μg/mL anti-MUC1, 80 µM tiliroside/5 μg/mL anti-MUC1, 160 µM tiliroside/5 μg/mL anti-MUC1). The stock solution of the tiliroside was 160 mM (prepared in DMSO; Sigma, St. Luis, MO, USA). In the next step, the cells were washed with PBS (Phosphate Buffered Saline, pH 7.4) and lysed for 20 min at 4 °C with RIPA buffer (Sigma, St. Luis, MO, USA) complemented with a cocktail of protease inhibitors (Sigma, St. Luis, MO, USA) diluted to 1:200 (in RIPA buffer). The collected lysates and culture media were centrifuged at 1000× *g* for 5 min at 4 °C. The obtained supernatants were frozen at −70 °C and used for Western blot and ELISA analyses. For quantitative Real-Time PCR assays, the monolayers were washed three times with sterile PBS (10 μM) and sonificed (Sonics Vibra cell; Sonics & Materials, Leicester-shire, UK) in order to disrupt the cell membranes. Aliquots of homogenate were used to isolate RNA. Cells without the added compounds were taken as control.

### 4.2. Cell Viability Test

The measurements of the viability of the cultured cells, in the presence of tiliroside and anti-MUC1, were determined as described previously [9,27] using 3-(4,5-dimethylthiazole-2-yl)-2,5-diphenyltetrazolium bromide (MTT) (Sigma, St. Louis, MO, USA) according to the procedure of Carmichael et al. [56].

### 4.3. RNA Isolation and Quantitative Real-Time PCR

Total RNA was isolated using Total RNA Mini Plus Concentrator (A&A Biotechnology, Gdańsk, Poland), according to the manufacturer’s instructions. The purity and concentration of RNA were assessed spectrophotometrically (Nanodrop 2000, Thermo Scientific, Waltham, MA, USA). Equal amounts (1 μg) of total RNA were subjected to reverse transcription using the SensiFAST^TM^ cDNA Synthesis Kit (Bioline, London, UK). A total of 20 μL of the reaction mixture contained RNA template, 1 μL of Reverse Transcriptase, 4 μL of 5×TransAmp Buffer, and DEPC-treated water. The conditions of incubation were 10 min at 25 °C, 30 min at 45 °C, and 5 min at 70 °C. The amplification of cDNA was performed using SensiFAST™ SYBR Kit (Bioline, London, UK) in the thermocycler CFX96 real-time system (BioRad, Hercules, CA, USA). The reaction mixture (20 μL) contained 2 μL of cDNA template (3-times diluted), 0.4 μL of each target-specific primer (10 μmol/L) (Genomed, Warsaw, Poland) (Table 1), 2×SensiFAST SYBR No-ROX Mix (5 μL), and DEPC-treated water. As a reference gene, glyceraldehyde-3-phosphate dehydrogenase (GAPDH) was used. The qRT-PCR parameters were as follows: 95 °C for 2 min (activation of DNA polymerase), followed by 40 cycles of 10 s at 95 °C (denaturation), 15 s at 60 °C (annealing), and 20 s at 72 °C (elongation). Each sample was examined in triplicate. The formation of the reaction products was confirmed by analysis of their melting curves. The ∆∆Ct method was applied to normalize the levels of target gene transcripts to GAPDH.

### 4.4. ELISA Tests

ELISA tests were applied to express the relative levels of sugar antigens: Tn, T, sialyl Tn, sialyl T, and Fucα1-3GlcNAc in cell lysates and the culture medium. The tests utilized specific lectins (Vector, Burlingame, CA, USA) with high affinities to the determined antigens. The following lectins were applied: VVA (*Vicia villosa* agglutinin) with specificity to Tn antigen (GalNAcα1-O-Ser/Thr), PNA (*Arachis hypogaea* agglutinin (peanut)) for T antigen (Galβ1,3-GalNAcα1-O-Ser/Thr), MAAII (*Maackia amurensis* agglutinin) for sialyl T, and LTA (*Lotus tetragonolobus* agglutinin) for Fucα1,3-GlcNAc structure. Microtiter plates (NUNC F96 Maxisorp, Roskilde, Denmark) were coated with 50 μL of cell lysates or culture medium (100 μg protein/mL) and incubated overnight at RT (room temperature). Next, there was the blocking step with 100 μL of blocking reagent for ELISA (Roche Diagnostics, Mannheim, Germany) for 1 h at RT. PBS with 0.05% Tween (100 μL, 3 times) was used in the washing of the wells. Then there was incubation with the respective lectins (5 μg/mL) for 2 h at RT. Next, 100 μL of horseradish peroxidase avidin D (Vector, Burlingame, CA, USA) was applied to detect carbohydrate antigens. After 30 min–1 h of incubation (RT), 100 μL of ABTS (2,2′-azino-bis(3-ethylbenzthiazoline-6-sulfonic acid) (Sigma, St. Luis, MO, USA) was used to develop the colored reaction. Absorbances were read at 405 nm after 20–40 min of incubation at RT. The samples were analyzed in triplicate. Instead of the samples and washing buffer, 1% BSA of lectins were applied as negative controls.

### 4.5. Western Blotting

Electrophoresis on 7.5–13% polyacrylamide gel and Western blotting analysis were performed to detect the expression of MUC1, C1GalT1, ST3Gal-IV, FUT4, Gal-3, pAkt, and NF-κB in cell lysates as well as MUC1 extracellular domain and Gal-3 in the culture media. The samples (20 μg of protein) were diluted in a probe sample buffer containing 2.5% SDS (Sigma, St. Luis, MO, USA), subjected to electrophoresis, and transferred to an Immobilon P membrane (Millipore, Bedford, MA, USA) according to the procedure of Towbin et al. [57]. To block the membranes, 5% skim milk in Tris Buffered Saline, pH 7.4 (TBS) with 0.05% Tween 20 (Sigma, St. Luis, MO, USA) was applied (1 h at RT). After the washing step in TBS-T, the membranes were incubated with proper primary monoclonal antibodies (Table 2) overnight at 4 °C. As negative controls, TBS-T buffer instead of antibodies were applied. To detect the immunoreactive complexes, secondary horseradish peroxidase-conjugated antibodies were used. The protein bands were visualized by enhanced chemiluminescence with Westar Supernova, ECL substrate for Western blotting (Cyangen, Bologna, Italy). The Gene Tools program (Syngene, Frederick, MD, USA) was applied to densitometrically quantify the intensity of the bands, which were normalized for β-actin.

### 4.6. Statistical Analysis

The results are presented as mean ± SD (standard deviation) from at least three independent determinations. Analyses were conducted using the Statistica package (StatSoft, Tulusa, OK, USA). One-way ANOVA followed by Duncan’s multiple post-hock test was applied to assess the statistical differences. Data with *p* > 0.05 were considered significant.

## 5. Conclusions

The proposed therapy utilizing anti-MUC1 with tiliroside seems to be a promising anti-gastric cancer therapy. We presented that the combined action of mAb and flavonoid was especially effective toward *C1GalT1*, *ST3GalT1*, *FUT4*, *Gal-3*, *NF-κB*, *Akt*, Tn antigen, T antigen, sialyl T antigen, and NF-κB expression. All these factors are inherently involved in cancer development, and we postulate that their inhibition can be an encouraging approach. However, we are aware of some limitations of our work. There are points that should be proved or explored more deeply in future experiments, e.g., in vivo studies, cell cycle analysis, or immunohistochemical studies.

## Figures and Tables

**Figure 1 ijms-24-13036-f001:**
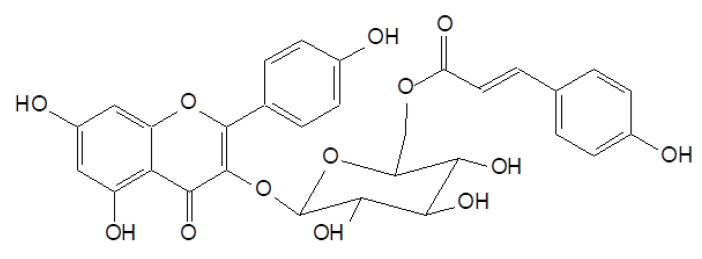
The structure of tiliroside.

**Figure 2 ijms-24-13036-f002:**
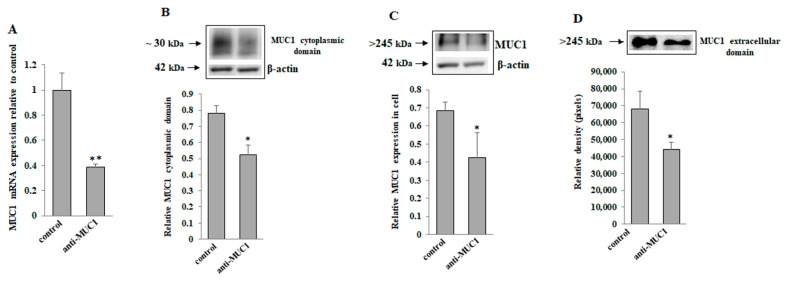
The effect of anti-MUC1 monoclonal antibody on MUC1 mRNA, MUC1 cytoplasmic domain, MUC1 in cell lysates, and extracellular domain in culture medium. AGS gastric cancer cells were incubated for 24 h with 5 µg/mL concentration of anti-MUC1 mAb. mRNA was assessed by quantitative Real-Time PCR (**A**). The results are presented as a relative fold change in mRNA expression of the gene in comparison with the gene in untreated control, where expression was set at 1. ±SD are the mean of triplicate cultures. ** *p* < 0.01. The cytoplasmic domain of MUC1 (**B**), MUC1 in cell lysates (**C**), and MUC1 in culture medium (**D**) was determined by Western blotting analysis. As a protein loading control, β-actin was applied. The bands’ intensities were quantified by densitometric studies. Data represent the mean ± SD of triplicate cultures. * *p* < 0.05 compared to control.

**Figure 3 ijms-24-13036-f003:**
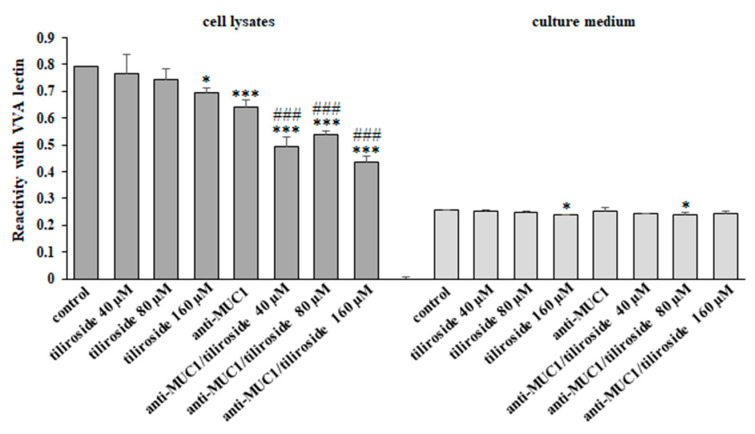
The effect of tiliroside and anti-MUC1 Tn antigen expression in cell lysates and the culture medium. The cells were incubated for 24 h with 40, 80, and 160 µM tiliroside, 5 µg/mL anti-MUC1, tiliroside + anti-MUC1. The relative Tn antigen expression in cell lysates and culture medium was assessed by ELISA test with biotinylated VVA lectin. The results are demonstrated as absorbance at 405 nm after reactivity with lectin. Values ± SD are the mean from three independent assays. * *p* < 0.05, *** *p* < 0.001 compared to control. ^###^ *p* < 0.001 compared to tiliroside monotherapy.

**Figure 4 ijms-24-13036-f004:**
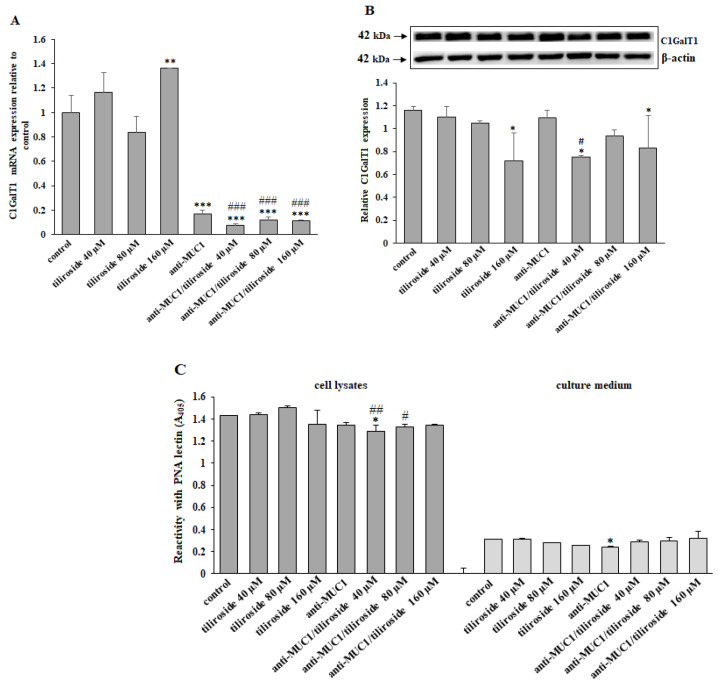
The effect of tiliroside and anti-MUC1 on C1GalT1 mRNA, C1GalT1 protein, and T antigen expression in cell lysates and culture medium. The cells were incubated for 24 h with 40, 80, and 160 µM tiliroside, 5 µg/mL anti-MUC1, tiliroside + anti-MUC1. C1GalT1 mRNA was assessed by quantitative Real-Time PCR (**A**). The results are presented as a relative fold change in the mRNA expression of the gene in comparison to the gene in the untreated control, where expression was set at 1. ±SD are the mean of triplicate cultures. ** *p* < 0.01, *** *p* < 0.001 compared to control. ^###^ *p* < 0.001 compared to tiliroside monotherapy. The C1GalT1 protein expression was determined by Western blotting (**B**). β-actin was applied as a protein loading control. The bands’ intensity was assessed by densitometric analysis. Data represent the mean ± SD of triplicate culture. * *p* < 0.05 compared to control. ^#^ *p* < 0.05 compared to tiliroside monotherapy. The relative T antigen expression in cell lysates and culture medium was assessed by ELISA test with biotinylated PNA lectin (**C**). The results are demonstrated as absorbance at 405 nm after reactivity with lectin. Values ± SD are the means from three independent assays. * *p* < 0.05 compared to control. ^#^ *p* < 0.05, ^##^ *p* < 0.01 compared to tiliroside monotherapy.

**Figure 5 ijms-24-13036-f005:**
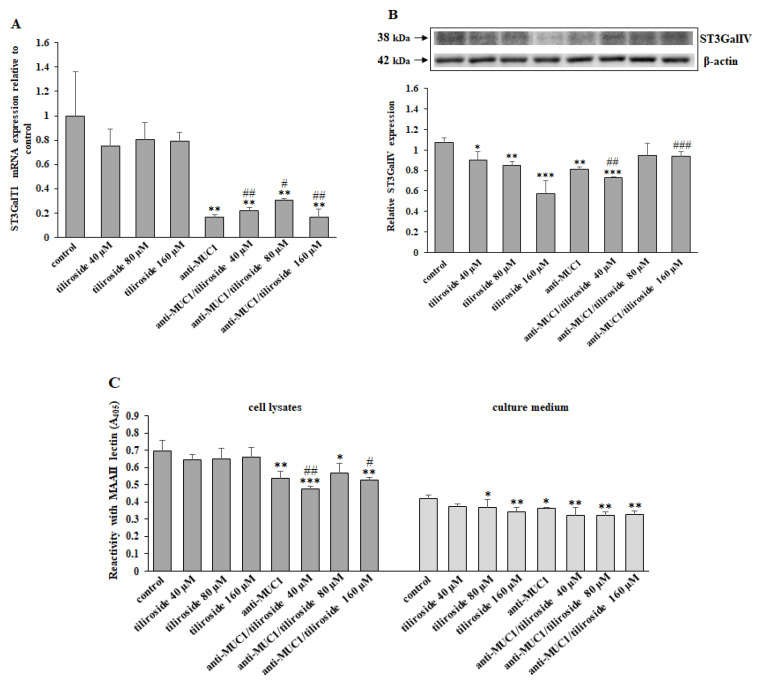
The effect of tiliroside and anti-MUC1 on ST3GalT1 mRNA, ST3GalIV protein, and sialyl T antigen expression in cell lysates and the culture medium. The cells were incubated for 24 h with 40, 80, and 160 µM tiliroside, 5 µg/mL anti-MUC1, tiliroside + anti-MUC1. ST3GalT1 mRNA was assessed by quantitative Real-Time PCR (**A**). The results are presented as a relative fold change in the mRNA expression of the gene in comparison with the gene in the untreated control, where expression was set at 1. ±SD are the mean of triplicate cultures. ** *p* < 0.01 compared to control. ^#^ *p* < 0.05, ^##^ *p* < 0.01 compared to tiliroside monotherapy. ST3GalTIV protein expression was determined by Western blotting (**B**). β-actin was applied as a protein loading control. The bands’ intensity was assessed by densitometric analysis. Data represent the mean ± SD of triplicate culture. * *p* < 0.05, ** *p* < 0.01, *** *p* < 0.001 compared to control. ^##^ *p* < 0.01, ^###^ *p* < 0.001 compared to tiliroside monotherapy. Relative sialyl T antigen expression in cell lysates and culture medium was assessed by ELISA test with biotinylated MAAII lectin (**C**). The results are demonstrated as absorbance at 405 nm after reactivity with lectin. Values ± SD are the mean from three independent assays. * *p* < 0.05, ** *p* < 0.01, *** *p* < 0.001 compared to control. ^#^ *p* < 0.05, ^##^ *p* < 0.01 compared to tiliroside monotherapy.

**Figure 6 ijms-24-13036-f006:**
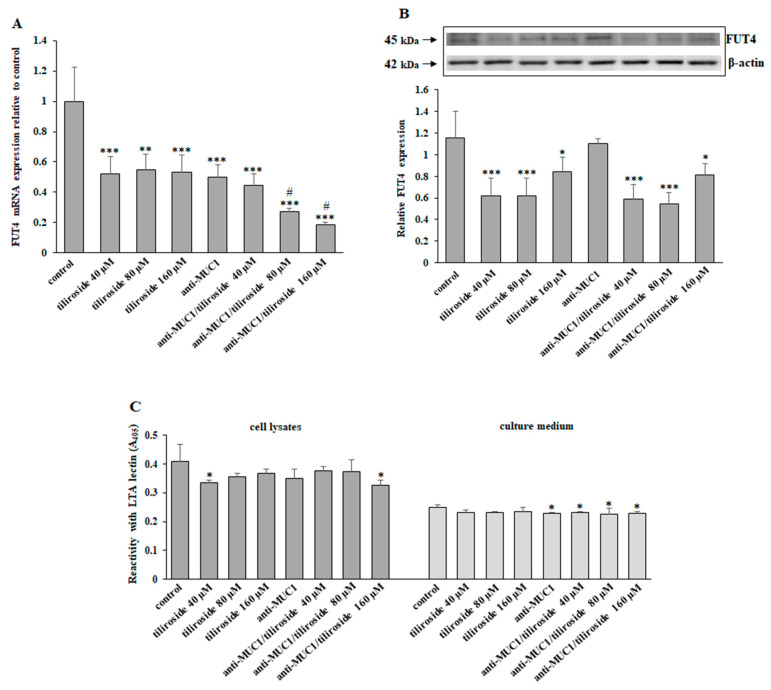
The effect of tiliroside and anti-MUC1 on FUT4 mRNA, FUT4 protein, and fucosylated (Fucα1,3-GalNAc) antigen expression in cell lysates and the culture medium. The cells were incubated for 24 h with 40, 80, and 160 µM tiliroside, 5 µg/mL anti-MUC1, tiliroside + anti-MUC1. FUT4 mRNA was assessed by quantitative Real-Time PCR (**A**). The results are presented as a relative fold change in the mRNA expression of the gene in comparison with the gene in the untreated control, where expression was set at 1. ±SD are the mean of triplicate cultures. ** *p* < 0.01, *** *p* < 0.001 compared to control. ^#^ *p* < 0.05 compared to tiliroside monotherapy. FUT4 protein expression was determined by Western blotting (**B**). β-actin was applied as a protein loading control. The bands’ intensity was assessed by densitometric analysis. Data represent the mean ± SD of triplicate culture. * *p* < 0.05, *** *p* < 0.001 compared to control. The relative Fucα1,3-GalNAc antigen expression in cell lysates and culture medium was assessed by ELISA test with biotinylated LTA lectin (**C**). The results are demonstrated as absorbance at 405 nm after reactivity with lectin. Values ± SD are the mean from three independent assays. * *p* < 0.05 compared to control.

**Figure 7 ijms-24-13036-f007:**
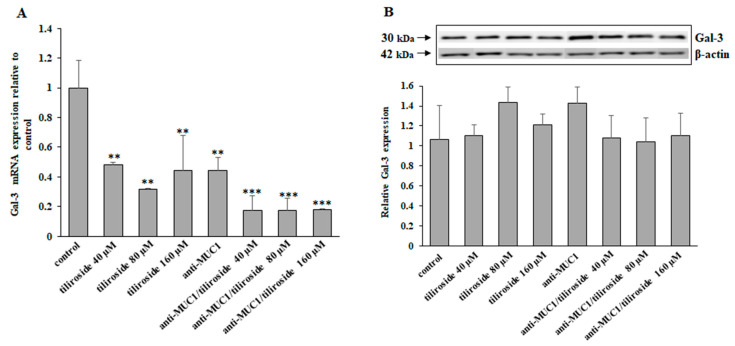
The effect of tiliroside and anti-MUC1 on Gal-3 mRNA and Gal-3 protein expression in cell lysates. The cells were incubated for 24 h with 40, 80, and 160 µM tiliroside, 5 µg/mL anti-MUC1, tiliroside + anti-MUC1. Gal-3 mRNA was assessed by quantitative Real-Time PCR (**A**). The results are presented as a relative fold change in the mRNA expression of the gene in comparison with the gene in the untreated control, where expression was set at 1. ±SD are the mean of triplicate cultures. ** *p* < 0.01, *** *p* < 0.001 compared to control. Gal-3 protein expression was determined by Western blotting (**B**). β-actin was applied as a protein loading control. The bands’ intensity was assessed by densitometric analysis. Data represent the mean ± SD of triplicate culture.

**Figure 8 ijms-24-13036-f008:**
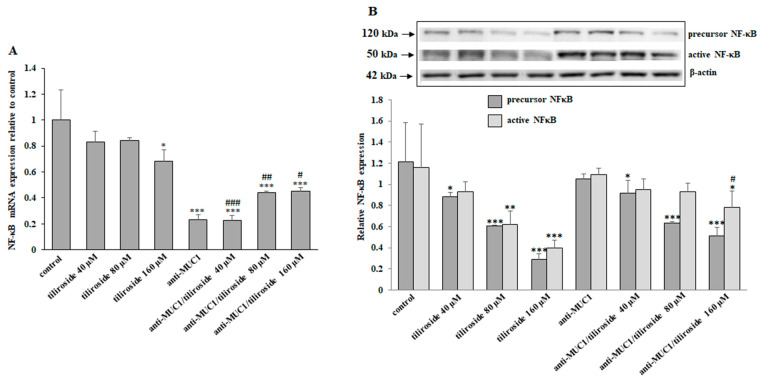
The effect of tiliroside and anti-MUC1 on NF-κB mRNA and NF-κB protein expression in cell lysates. The cells were incubated for 24 h with 40, 80, and 160 µM tiliroside, 5 µg/mL anti-MUC1, tiliroside + anti-MUC1. NF-κB mRNA was assessed by quantitative Real-Time PCR (**A**). The results are presented as a relative fold change in the mRNA expression of the gene in comparison with the gene in the untreated control, where expression was set at 1. ±SD are the mean of triplicate cultures. * *p* < 0.05, *** *p* < 0.001 compared to control. ^#^ *p* < 0.05, ^##^
*p* < 0.01, ^###^ *p* < 0.001 compared to tiliroside monotherapy. NF-κB protein expression was determined by Western blotting (**B**). β-actin was applied as a protein loading control. The bands’ intensity was assessed by densitometric analysis. Data represent the mean ± SD of triplicate cultures. * *p* < 0.05, ** *p* < 0.01, *** *p* < 0.001 compared to control. # *p* < 0.05 compared to tiliroside monotherapy.

**Figure 9 ijms-24-13036-f009:**
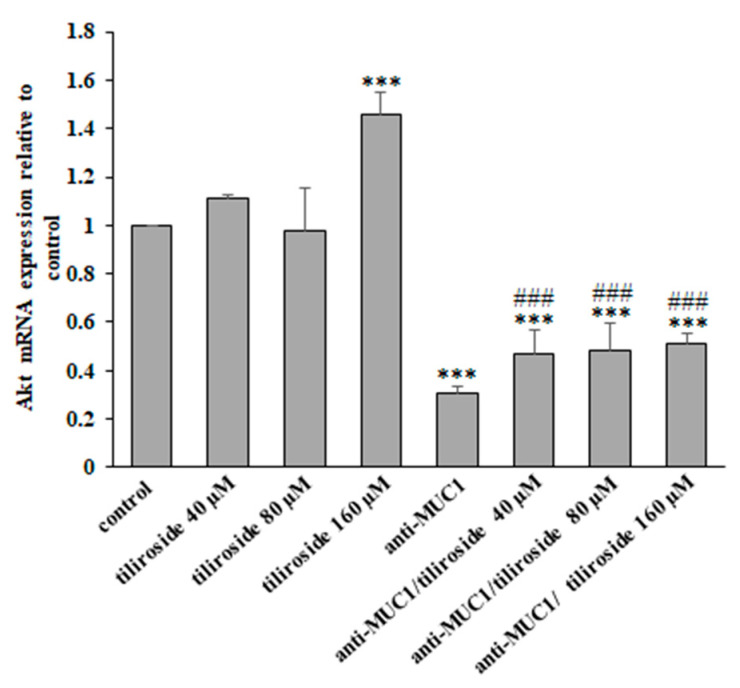
The effect of tiliroside and anti-MUC1 on Akt mRNA. The cells were incubated for 24 h with 40, 80, and 160 µM tiliroside, 5 µg/mL anti-MUC1, tiliroside + anti-MUC1. mRNA was assessed by quantitative Real-Time PCR. The results are presented as a relative fold change in the mRNA expression of the gene in comparison with the gene in the untreated control, where expression was set at 1. ±SD are the mean of triplicate cultures. *** *p* < 0.001 compared to control; ### *p* < 0.001 compared to tiliroside monotherapy.

**Table 1 ijms-24-13036-t001:** Primers used for quantitative Real-Time PCR.

Gene	Forward Primer (5′ → 3′)	Reverse Primer (5′ → 3′)
*MUC1*	TGCCTTGGCTGTCTGTCAGT	GTAGGTATCCCGGGCTGGAA
*C1GalT1*	AAGCAGGGCTACATGAGTGG	GCATCTCCCCAGTGCTAAGT
*ST3GalT1*	TCGGCCTGGTTCGATGA	CGCGTTCTGGGCAGTCA
*FUT4*	AAGCCGTTGAGGCGGTTT	ACAGTTGTGTATGAGATTTGGAAGCT
*Gal-3*	GCAGACAATTTTTCGCTCCATG	CTGTTGTTCTCATTGAAGCGTG
*Akt*	TCTATGGCGCTGAGATTGTG	CTTAATGTGCCCGTCCTTGT
*NF-κB*	CTGAACCAGGGCATACCTGT	GAGAAGTCCATGTCCGCAAT
*GAPDH*	GTGAACCATGAGAAGTATGACAA	CATGAGTCCTTCCACGATAC

**Table 2 ijms-24-13036-t002:** Antibodies used in Western blotting.

Antibody	Clone	Source
Anti-MUC1; extracellular domain (mouse IgG)	BC2	Abcam
Anti-MUC1; cytoplasmic tail (Armenian hamster IgG)	CT2	Abcam
Anti-NF-κB (mouse IgG)	5D10D11	Cell Sign Tech
Anti-C1GalT1 (mouse IgG)	F-31	Santa Cruz
Anti-FUT4 (mouse IgG)	A-10	Santa Cruz
Anti-ST3GalIV (mouse IgG)	1F4	Santa Cruz
Anti-Gal-3 (mouse IgG)	B2C10	Santa Cruz
Anti-β-actin (rabbit IgG)		Sigma
Anti-mouse IgG peroxidase conjugated		Sigma
Anti-rabbit IgG peroxidase conjugated		Sigma
Anti-armenian hamster IgG peroxidase conjugated		Abcam

## Data Availability

Data available on request.

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
