# Peer review of "Tiliroside Combined with Anti-MUC1 Monoclonal Antibody as Promising Anti-Cancer Strategy in AGS Cancer Cells"

_ijms, 2023, doi:10.3390/ijms241713036_

Round 1

Reviewer 1 Report

In this manuscript, Radziejewska et al. analyze how tiliroside and anti-MUC1 antibody, separately or in combination, affect levels of specific glycosylation modifications and the expression of key enzymes responsible for producing these modifications. The authors focused on tumor associated carbohydrate antigens, using AGS gastric cancer cells to address the therapeutic value of tiliroside and anti-MUC1 antibody together. They also examined the effects of tiliroside and anti-MUC1 antibody on NF-kB activation and galectin-3 and Akt expression, which are generally associated with cancer. Overall, the authors found tiliroside and anti-MUC1 caused a moderate reduction in expression of the glycosyltransferases and the carbohydrate antigens they produce, which is interesting in the pursuit of finding natural products with anti-cancer properties. Given that MUC1 glycosylation appeared to be an important point to this study, analysis of MUC1 glycosylation in cell lysates and the extracellular medium would have added value to this study. There are some concerns with the experimental results that are included with the following specific points: 

In the Introduction, please include a paragraph describing gastric cancer. 

It is concerning that there appears to be little relationship between the real-time PCR and the western blot results (figures 4, 5, 7 and 8). A quick analysis of ppGalNAcT2 (GALNT2) revealing that the amplicon is >600bp, which is extremely large for RT/real-time analysis. Based on this concern, were amplification efficiencies determined for the PCR primer sets? If so, please include these data in the Materials and Methods. 

Figure 2: For quantification of the western blots, why is a different axis used for 2D (pixel density) compared to 2B and C? Please explain the relevance/role of the MUC1 cytoplasmic domain and the secreted MUC1 extracellular domain. Please state the specimen used for the MUC1 cytoplasmic domain western (assume it’s cell lysate; also assume the MUC1 extracellular domain antibody was used to analyze expression in cells. 

For the ELISAs, the figure legend states relative antigen expression was assessed; however, values appear to be for absorbance. Please clarify this point. 

In the Discussion, please speculate on how the treatments can affect mRNA but not protein levels (e.g., figure 4). 

In the Discussion, it is speculated that the MUC1 antibody reduces MUC1 expression by triggering MUC1 internalization. Is MUC1 then believed to traffic to lysosomes for degradation? 

In the Discussion, there are statements regarding how aberrant glycosylation affects cancer cell phenotypes and immune evasion. For each of these statements, please state the specific cancer for these reported effects as they might not generally apply to all types of cancer. 

From the western blot shown in figure 8D, it appears that the anti-MUC1 antibody actually promotes NF-kB activation, while tiliroside causes a general reduction in NF-KB (precursor and activated; disregarded the mRNA data because the protein data is relevant to NF-kB signaling). However, the authors imply that the combination of tiliroside and MUC1 antibody inhibit NF-kB, which doesn’t seem to be consistent with the data. Again, the protein and not the mRNA is relevant to signaling. 

The English language was good. While there were some grammatical errors, the manuscript was easy to read.

Author Response

Thanks for very much for valuable comments and suggestions concerning our experiments. They are really very important for us in improving a current work and they will be also taken into account in our future studies that will be continued on the subject. We have tried to introduced most of the suggested changes.

We hope that we satisfied most of the reviewers’ expectations.

All the changes are seen in red with specific comments.

Response to Reviewer 1:

  1. In the Introduction, please include a paragraph describing gastric cancer - Some words about gastric cancer have been added to Introduction section, two new references have been added;

  1. It is concerning that there appears to be little relationship between the real-time PCR and the western blot results (figures 4, 5, 7 and 8) – We agree that in some cases mRNA doesn’t correlate with protein. There can be some reasons of such results. E.g. the protein could be regulated post-translationally; there can be many regulatory steps between mRNA and protein; there can be different half-lives of mRNA and protein, cleavage and degradation of protein…; it is also possible to have low mRNA expression and high level of protein if the protein directly influences its expression (a negative feedback);

A quick analysis of ppGalNAcT2 (GALNT2) revealing that the amplicon is >600bp, which is extremely large for RT/real-time analysis. Based on this concern, were amplification efficiencies determined for the PCR primer sets? If so, please include these data in the Materials and Methods – We absolutely agree with this remark. We agree that we used improper primer. It was our mistake. Thus because of this we decided to remove this result. That is due to lack of material to repeat the determination of  ppGalNAcT2 mRNA expression with proper primer. We absolutely take this into consideration in our future experiments;

  1. Figure 2: For quantification of the western blots, why is a different axis used for 2D (pixel density) compared to 2B and C? Please explain the relevance/role of the MUC1 cytoplasmic domain and the secreted MUC1 extracellular domain. Please state the specimen used for the MUC1 cytoplasmic domain western (assume it’s cell lysate; also assume the MUC1 extracellular domain antibody was used to analyze expression in cells - In Figures 2B and 2C cell lysates were analyzed, the same amount of proteins were applied on the gels, and intensities of the bands were normalized to β-actin; in case of Figure 2D, culture medium was analyzed, also the same amount of proteins were applied, but of course the intensities of the bands couldn’t be normalized to β-actin, so pixels density was used.

Some words about roles of cytoplasmic domain and extracellular domain were added in Introduction section;

  1. For the ELISAs, the figure legend states relative antigen expression was assessed; however, values appear to be for absorbance. Please clarify this point – We typically use the results for ELISAs with absorbance values when we don’t have proper standard to have results in proper unit. For analyzed antigens we don’t have standards so we assess only the “relative’ changes in accordance with untreated control;

  1. In the Discussion, please speculate on how the treatments can affect mRNA but not protein levels (e.g., figure 4) – The explanation can be found in the answer to point 2 (above). Some words were added in Discussion section;

  1. In the Discussion, it is speculated that the MUC1 antibody reduces MUC1 expression by triggering MUC1 internalization. Is MUC1 then believed to traffic to lysosomes for degradation? – This notion is based on the literature data. Upon, on our results we can only speculate that MUC1 (in some part) could be degraded in lysosomes;

  1. In the Discussion, there are statements regarding how aberrant glycosylation affects cancer cell phenotypes and immune evasion. For each of these statements, please state the specific cancer for these reported effects as they might not generally apply to all types of cancer – Added according to suggestions;

  1. From the western blot shown in figure 8D, it appears that the anti-MUC1 antibody actually promotes NF-kB activation, while tiliroside causes a general reduction in NF-KB (precursor and activated; disregarded the mRNA data because the protein data is relevant to NF-kB signaling). However, the authors imply that the combination of tiliroside and MUC1 antibody inhibit NF-kB, which doesn’t seem to be consistent with the data. Again, the protein and not the mRNA is relevant to signaling – Again we agree that not always mRNA correlates with protein expression (point 2 above). According to results presented in Figure 8B we assume that anti-MUC1 didn’t influence NF-κB protein expression (both forms). And we agree that combined action didn’t enhance the effect NF-κB protein expression. Proper short explanations were introduced to the text.

Reviewer 2 Report

The authors of this manuscript have looked at combined effect of tiliroside with anti-MUC1 monoclonal antibody as potential anti-cancer therapy for AGS cancer cells. They used various methods including real time PCR, ELISA etc. to verify the MUC1 expression and expression of various tumor associated antigens. They conclude that combined action was effective in reducing the level of many tumor associated antigens and hence indicating that this combined therapy may be a promising anti-gastric cancer treatment strategy. However, there are some minor text editing required to make the manuscript better. The authors should also do a couple of experiments to strengthen the manuscript.

Line 77: Section 2.1 can go in materials and methods section or in supplementary section as it is not adding anything substantial in the results section.

Figure 2D. are the bands normalized to any control? Is there a ponceau stained blot, to verify equal loading?

Figure 3B (culture medium): It would be great if there could be an explanation to why there is not any significant changes in level of Tn antigen in the culture medium after treatment?

The authors look at the inhibition of ppGALNT2, however other transferases such as ppGALNT1/ ppGALNT3/ ppGALNT4 can also initiate MUC1 glycosylation. I was wondering if the authors have looked at inhibition of these GALNT’s as well with the tiliroside and anti-MUC1 antibody.

I would suggest authors to perform some ELISA/WB to look at the levels of these other GALNT’s  (NT1/NT3/NT4) in the cells treated with tiliroside, anti-MUC1 antibody, and combination of both. It would be great to see if this combination also affects other GALNT’s involved in initiating the glycosylation on MUC1.

Line 527: Proper lectinsà respective lectins.

Line 87-89: Reframe the sentence so that it’s easier to understand.

Line 536-538: Reframe the sentence so that it’s easier to understand.

Line 547: Protein bondà protein band.

Line 560: Presenting therapyà Proposed therapy.

In supplementary figure S5B and S8B, there is no legend. Include the name of the target protein being identified (ST3GalIV or NF-kB), like other figures.

Some of the sentences are convoluted and needs to reframed, to make it understandable.

I have included the comments in the general comments and suggestions for authors section.

Author Response

Thanks for very much for valuable comments and suggestions concerning our experiments. They are really very important for us in improving a current work and they will be also taken into account in our future studies that will be continued on the subject. We have tried to introduced most of the suggested changes.

We hope that we satisfied most of the reviewers’ expectations.

All the changes are seen in red with specific comments.

Response to Reviewer 2:

  1. Line 77: Section 2.1 can go in materials and methods section or in supplementary section as it is not adding anything substantial in the results section – Generally we agree. However in this short section there is information about proper concentrations of applied compounds and because of this we placed this in Results section;

  1. Figure 2D. are the bands normalized to any control? Is there a ponceau stained blot, to verify equal loading? – Routinely we don’t use a ponceau stained blot (sometimes we used but analysis were problematic) because the same amount of proteins were applied, but of course the intensities of the bands couldn’t be normalized to β-actin, so pixels density was used.

  1. Figure 3B (culture medium): It would be great if there could be an explanation to why there is not any significant changes in level of Tn antigen in the culture medium after treatment? – Truly it is difficult to explain lack of clear changes in the level of this antigen in culture medium (very small changes are seen only after action of 160 µM tiliroside and 80 µM tiliroside and mAb). We can only speculate that e.g. the expression of Tn antigen could be too low to be detected by lectin in ELISA test;

  1. The authors look at the inhibition of ppGALNT2, however other transferases such as ppGALNT1/ ppGALNT3/ ppGALNT4 can also initiate MUC1 glycosylation. I was wondering if the authors have looked at inhibition of these GALNT’s as well with the tiliroside and anti-MUC1 antibody – Thank you for this valuable comment. Unfortunately in this work we didn’t check the expression of other transferases. Currently we don’t have samples to perform such analysis. However such valuable suggestion will be taken into account in our future experiments;

  1. I would suggest authors to perform some ELISA/WB to look at the levels of these other GALNT’s (NT1/NT3/NT4) in the cells treated with tiliroside, anti-MUC1 antibody, and combination of both. It would be great to see if this combination also affects other GALNT’s involved in initiating the glycosylation on MUC1 – See above

  1. Line 527: Proper lectinsà respective lectins - Corrected according to suggestions;

  1. Line 87-89: Reframe the sentence so that it’s easier to understand – Corrected according to suggestions;

  1. Line 536-538: Reframe the sentence so that it’s easier to understand – Corrected according to suggestions;

  1. Line 547: Protein bondà protein band - Corrected according to suggestion;

  1. Line 560: Presenting therapyà Proposed therapy - Corrected according to suggestions;

  1. In supplementary figure S5B and S8B, there is no legend. Include the name of the target protein being identified (ST3GalIV or NF-kB), like other figure - Proper legends have been added.

Round 2

Reviewer 2 Report

The authors of this manuscript have looked at combined effect of tiliroside with anti-MUC1 monoclonal antibody as potential anti-cancer therapy for AGS cancer cells. They used various methods including real time PCR, ELISA etc., to verify the MUC1 expression and expression of various tumor associated antigens. They conclude that combined action was effective in reducing the level of many tumors associated antigens and hence indicating that this combined therapy may be a promising anti-gastric cancer treatment strategy. I would like to thank the authors to address majority of the comments and suggestions.  It is understandable that some experiments are hard to do with limited samples.